# Systematically optimized BCMA/CS1 bispecific CAR-T cells robustly control heterogeneous multiple myeloma

Eugenia Zah[1,6], Eunwoo Nam [1], Vinya Bhuvan[1], Uyen Tran[2], Brenda Y. Ji[1], Stanley B. Gosliner[1], Xiuli Wang[3], Christine E. Brown[3] & Yvonne Y. Chen [1,4,5✉]

Chimeric antigen receptor (CAR)-T cell therapy has shown remarkable clinical efficacy against B-cell malignancies, yet marked vulnerability to antigen escape and tumor relapse exists. Here we report the rational design and optimization of bispecific CAR-T cells with robust activity against heterogeneous multiple myeloma (MM) that is resistant to conventional CAR-T cell therapy targeting B-cell maturation antigen (BCMA). We demonstrate that BCMA/CS1 bispecific CAR-T cells exhibit superior CAR expression and function compared to T cells that co-express individual BCMA and CS1 CARs. Combination therapy with anti–PD-1 antibody further accelerates the rate of initial tumor clearance in vivo, while CAR-T cell treatment alone achieves durable tumor-free survival even upon tumor re-challenge. Taken together, the BCMA/CS1 bispecific CAR presents a promising treatment approach to prevent antigen escape in CAR-T cell therapy against MM, and the vertically integrated optimization process can be used to develop robust cell-based therapy against novel disease targets.

[1] Department of Chemical and Biomolecular Engineering, University of California–Los Angeles, 420 Westwood Plaza, BH 5513 Los Angeles, CA, USA.
[2] Department of Chemistry and Biochemistry, University of California–Los Angeles, 420 Westwood Plaza, BH 5513 Los Angeles, CA, USA. [3] Department of Hematology and Hematopoietic Cell Transplantation, T Cell Therapeutics Research Laboratory, City of Hope Beckman Research Institute and Medical Center, 1500 E. Duarte Rd., Duarte, CA, USA. [4] Department of Microbiology, Immunology, and Molecular Genetics, University of California–Los Angeles, 420 Westwood Plaza, BH 5513 Los Angeles, CA, USA. [5] Parker Institute for Cancer Immunotherapy Center at UCLA, 420 Westwood Plaza, BH 5513 Los Angeles, CA, USA. [6]Present address: Amgen, Thousand Oaks, CA, USA. ✉email: yvonne.chen@ucla.edu

Multiple myeloma (MM) is the second-most common hematologic malignancy, causing 98,437 deaths globally in 2016, with an estimated 32,110 new diagnoses in the US in 2019[1,2]. In recent years, immunomodulatory drugs and proteasome inhibitors, such as thalidomide, lenalidomide, and bortezomib, which may be administered in conjunction with autologous stem-cell transplant, have substantially improved survival of patients suffering from MM[3]. However, MM remains an incurable disease despite these therapeutic options.

The adoptive transfer of CAR-T cells targeting B-cell maturation antigen (BCMA) has shown clinical efficacy against MM, achieving 80–100% overall response rate across multiple clinical trials[4–8]. However, BCMA is not uniformly expressed on MM cells, as evidenced by a recent study that screened 85 MM patients and found 33 to be BCMA negative[4], thus limiting patient eligibility for BCMA CAR-T-cell therapy. Furthermore, multiple cases of patient relapse involving tumor cells with downregulated BCMA expression have been reported[4,5,7], underscoring antigen escape as a significant obstacle in the treatment of MM with BCMA CAR-T cell therapy. In addition, a substantial fraction of patients treated with BCMA CAR-T cells eventually relapse even when BCMA expression is retained[4,5,7], suggesting a lack of durable effector function by the engineered T cells.

To address these challenges, we set out to develop a new CAR-T cell treatment for MM exhibiting greater resistance to antigen escape and improved long-term effector function. As a living drug, CAR-T cells constitute a complex treatment modality involving multiple process parameters that extend well beyond the CAR molecule itself. Therefore, we developed a vertically integrated optimization process that begins with structure-guided design and high-throughput functional screening of CAR variants, followed by systematic identification of optimal cell-manufacturing conditions, and ending with the evaluation of long-term in vivo antitumor efficacy of CAR-T cell therapy alone and in combination with checkpoint inhibitor therapy (Fig. 1a).

We and others have previously demonstrated that T cells expressing single-chain bispecific CARs, also known as "tandem CARs," can prevent antigen escape and improve the efficacy of CAR-T cell therapy[9–12]. When optimized, these single-chain bispecific receptors can serve as "OR-gate" CARs that enable T cells to effectively target tumor cells that present either antigen A or antigen B, thus requiring tumor cells to lose both antigens before escaping T-cell detection. Toward the goal of dual-antigen targeting for MM, a CAR incorporating a truncated "a proliferation-inducing ligand" (dAPRIL) as the ligand-binding domain had previously been reported to interact with both BCMA and another MM marker, the transmembrane activator and CAML interactor (TACI)[13]. However, TACI, like BCMA, exhibits heterogeneous expression in MM, and MM cells lacking expression of both antigens have been previously identified[14–16]. In contrast, CS1 (also known as SLAMF7 or CD319) is highly expressed on multiple types of MM and has been found on 90–97% of patient MM samples[17,18], and an anti-CS1 CAR-T-cell therapy[19] is currently being tested in the clinic (NCT03710421). We reason that simultaneous targeting of BCMA and CS1 would leverage the therapeutic efficacy of BCMA targeting while providing a safeguard against tumor escape due to BCMA loss.

Here we report the rapid design and optimization of BCMA/CS1 OR-gate CAR-T cells that can efficiently target BCMA- or CS1-expressing tumor cells while maintaining robust ex vivo expansion with minimal fratricidal side effects. We show that BCMA/CS1 OR-gate CAR-T cells have superior CAR expression and proliferative capability compared to T cells co-expressing two separate CARs targeting BCMA and CS1. Furthermore, BCMA/CS1 OR-gate CAR-T cells are substantially more effective than single-input BCMA or CS1 CAR-T cells in controlling heterogeneous MM tumor populations in vivo, resulting in significantly prolonged survival of tumor-bearing mice. Finally, we demonstrate that combination therapy with anti-PD-1 antibody increases the speed of initial in vivo tumor clearance by BCMA/CS1 OR-gate CAR-T cells against established MM, but OR-gate CAR-T cells alone are sufficient to eradicate high tumor burdens, albeit over a longer time period, leading to effective and durable control of highly aggressive tumors.

## Results

**Construction of single-chain bispecific BCMA/CS1 CARs.** A panel of second-generation, 4-1BB–containing OR-gate CAR variants was constructed to evaluate multiple ligand-binding moieties, including three BCMA-recognition domains (dAPRIL and single-chain variable fragments (scFvs) derived from two BCMA-binding antibodies, c11D5.3 or J22.9-xi), each paired with one of two CS1-binding scFvs (Luc90 or huLuc63) (Fig. 1b). We and others have shown that optimal CAR signaling requires the CAR's ligand-binding domain to be precisely positioned to create an immunological synapse of an appropriate dimension when bound to the target antigen[10,20–22]. No epitope mapping data had been reported for the BCMA-binding domains considered in this study. However, the CS1-targeting antibodies huLuc63 and Luc90 are known to bind the membrane-proximal C2 epitope and the membrane-distal V epitope of CS1, respectively[23] (Supplementary Fig. 1a). Therefore, we reasoned that the huLuc63-derived scFv should be placed at the membrane-distal position relative to the T-cell membrane, paired with a BCMA-binding domain at the membrane-proximal position, such that the huLuc63 scFv can have sufficient extension to make proper contact with the C2 epitope close to the target-cell surface. Conversely, we fixed the Luc90 scFv at the membrane-proximal position for a second set of CARs (Supplementary Fig. 1b). To increase potential clinical applicability, both murine and humanized versions of the BCMA-binding c11D5.3 and J22.9-xi scFvs were evaluated. All OR-gate CARs in this initial panel contained a short (12-amino-acid) extracellular spacer. In total, ten bispecific CARs plus seven single-input CAR controls were constructed for the first round of screening (Fig. 1b, Supplementary Fig. S1b, c).

**Rapid functional testing of BCMA/CS1 OR-gate CAR designs.** A methodology for high-throughput generation and screening of new CAR-T cells was developed to support the rapid evaluation of novel OR-gate CAR designs (Fig. 1c; see Methods), with low-volume functional assays that enabled simultaneous comparison of up to 17 different T-cell lines, all generated using cells from the same donor to ensure comparability (Supplementary Fig. 2a).

CAR surface expression staining revealed that receptors comprising Luc90 paired with either humanized or murine J22.9-xi scFv were poorly expressed on primary human T cells and thus eliminated from further consideration (Supplementary Fig. 2). Among the remaining eight OR-gate candidates, c11D5.3-Luc90 and huc11D5.3-Luc90 CAR-T cells were the most effective against BCMA$^+$ target cells based on both target-cell lysis and antigen-stimulated T-cell proliferation (Fig. 2a, b; Supplementary Fig. 3a), whereas CS1 was best targeted by huLuc63-c11D5.3 (Fig. 2b).

The top five OR-gate CAR-T-cell lines based on target-cell lysis and T-cell proliferation (Fig. 2b, marked by arrows) were subjected to repeated antigen challenge to evaluate their propensity for exhaustion. CAR-T cells were challenged with BCMA$^+$/CS1$^+$ K562 cells in the first two rounds, followed by BCMA$^+$/CS1$^-$ and BCMA$^-$/CS1$^+$ K562 cells in the third and fourth rounds, respectively. Here, c11D5.3-Luc90, huc11D5.3-Luc90, and huLuc63-c11D5.3 CAR-T cells continued to

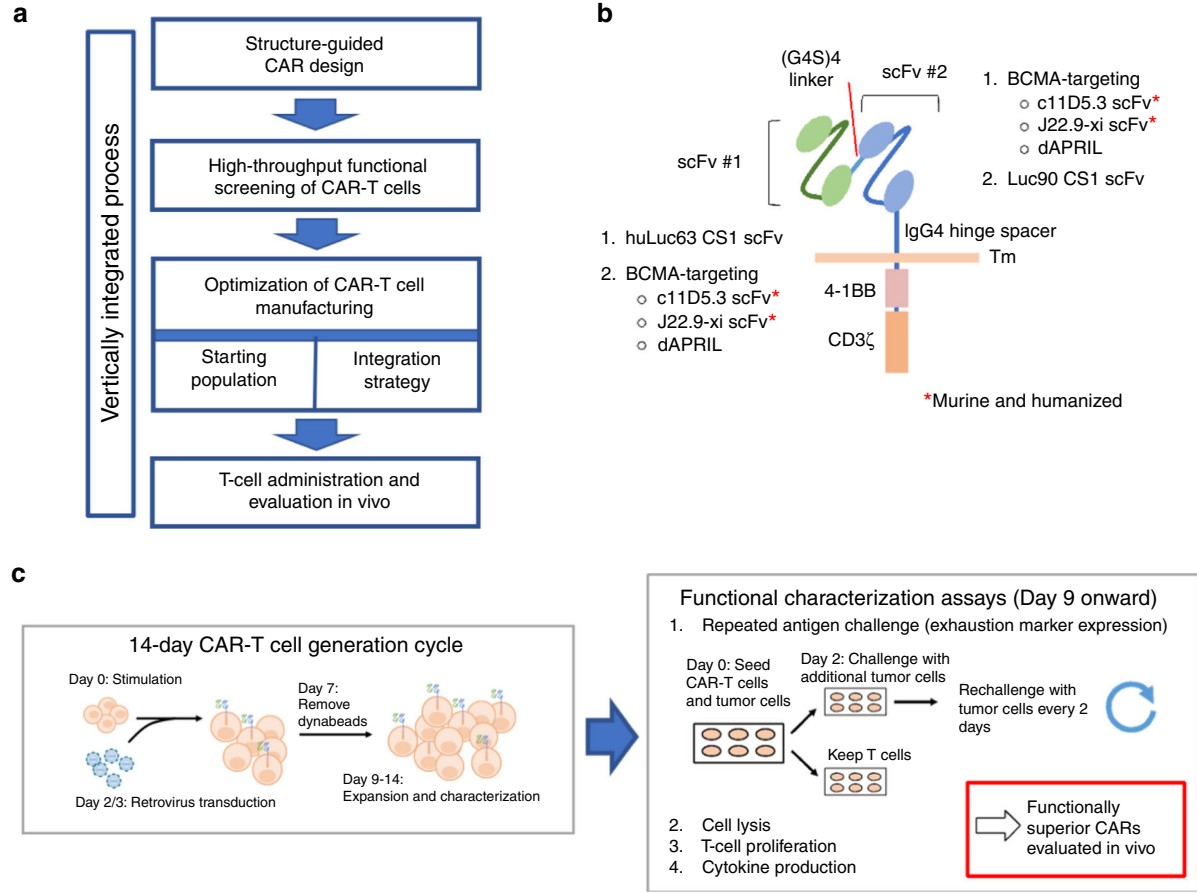

**Fig. 1 High-throughput generation and screening of BCMA/CS1 OR-gate CARs. a** A vertically integrated optimization process for CAR-T cell therapy development. **b** Schematic of a single-chain bispecific (OR-gate) CAR, which contains two ligand-binding domains connected in tandem. A panel of CAR variants was constructed from two CS1-binding scFvs and three BCMA-specific binding domains. Both murine and humanized versions of BCMA-binding scFvs (c11D5.3 and J22.9-xi) were evaluated. The CS1-binding huLuc63 and Luc90 scFvs were fixed at the membrane-distal and membrane-proximal positions, respectively, based on binding-epitope analysis. **c** Methodology for producing and screening bispecific CARs. CAR-T cells were generated in a 14-day cycle. Starting on day 9 post-stimulation, CAR-T cells were characterized for antitumor function in various assays, which could last up to 2 weeks.

outperform other candidates by maintaining efficient target-cell killing for three rounds of antigen challenge before succumbing to tumor outgrowth (Fig. 2c, Supplementary Fig. 3b). Antigen expression patterns on surviving tumor cells confirmed that c11D5.3-Luc90 and huc11D5.3-Luc90 CAR-T cells showed superior targeting of BCMA$^+$ tumor cells, resulting in a disproportionately large fraction of BCMA$^-$/CS1$^+$ K562 cells in the remaining tumor population (Fig. 2d, Supplementary Fig. 3b). In contrast, huLuc63-c11D5.3 CAR-T cells had a higher proportion of BCMA$^+$/CS1$^-$ target cells remaining, indicating greater efficacy against CS1$^+$ targets.

Surprisingly, all dAPRIL-based CARs failed to achieve efficient target-cell lysis and T-cell proliferation (Fig. 2a–c), and defective BCMA-targeting appeared to be the main cause based on the composition of residual tumor cells (Fig. 2d). Given these results, the dAPRIL-based designs evaluated in this panel were eliminated from further consideration.

**Antigen-recognition tradeoff by single-chain OR-Gate CARs.** In the repeated antigen challenge assay, we had observed that the single-input c11D5.3 BCMA CAR showed superior function when coupled to a long (229-amino-acid) extracellular spacer (Supplementary Fig. 3c). This observation is expected given that BCMA has a very short (36-amino-acid) ectodomain, thus the BCMA CAR needs to extend farther out to reach the target antigen. However, as previously noted, the binding epitope for the

CS1-targeting huLuc63 scFv is also expected to work best with a long spacer (Supplementary Fig. 1a), thus raising the prospect of an unavoidable tradeoff between BCMA and CS1 targeting. Indeed, when we evaluated the effect of lengthening of the extracellular spacer (from 12 to 229-amino acids) and/or changing the relative positioning of the two scFvs, we found the original huLuc63-c11D5.3 Short CAR design to possess the best balance of BCMA and CS1-targeting efficiency while requiring a relatively compact DNA footprint (Supplementary Figs. 4a and 5). Based on cumulative in vitro functional assay results, we chose to focus on huc11D5.3-Luc90 Short and huLuc63-c11D5.3 Short as our final two candidates, each with a slight advantage against BCMA or CS1, respectively.

**Superiority of single-chain OR-gate CARs over dual-CARs.** The fact that the single-chain bispecific CAR structure employed here could not be fully optimized for both BCMA and CS1 targeting due to overlapping and thus incompatible structural preferences for the two target epitopes raises the question of whether (a) co-expressing two separate single-input CARs ("DualCAR" approach)[24] or (b) coadministering two separate single-input CAR-T-cell products ("CARpool" approach) would be a more effective way to achieve T-cell bispecificity. The CARpool strategy is the costliest and most operationally complex approach, as it requires the production and infusion of two separate cell products per patient. Furthermore, the CARpool approach has been shown

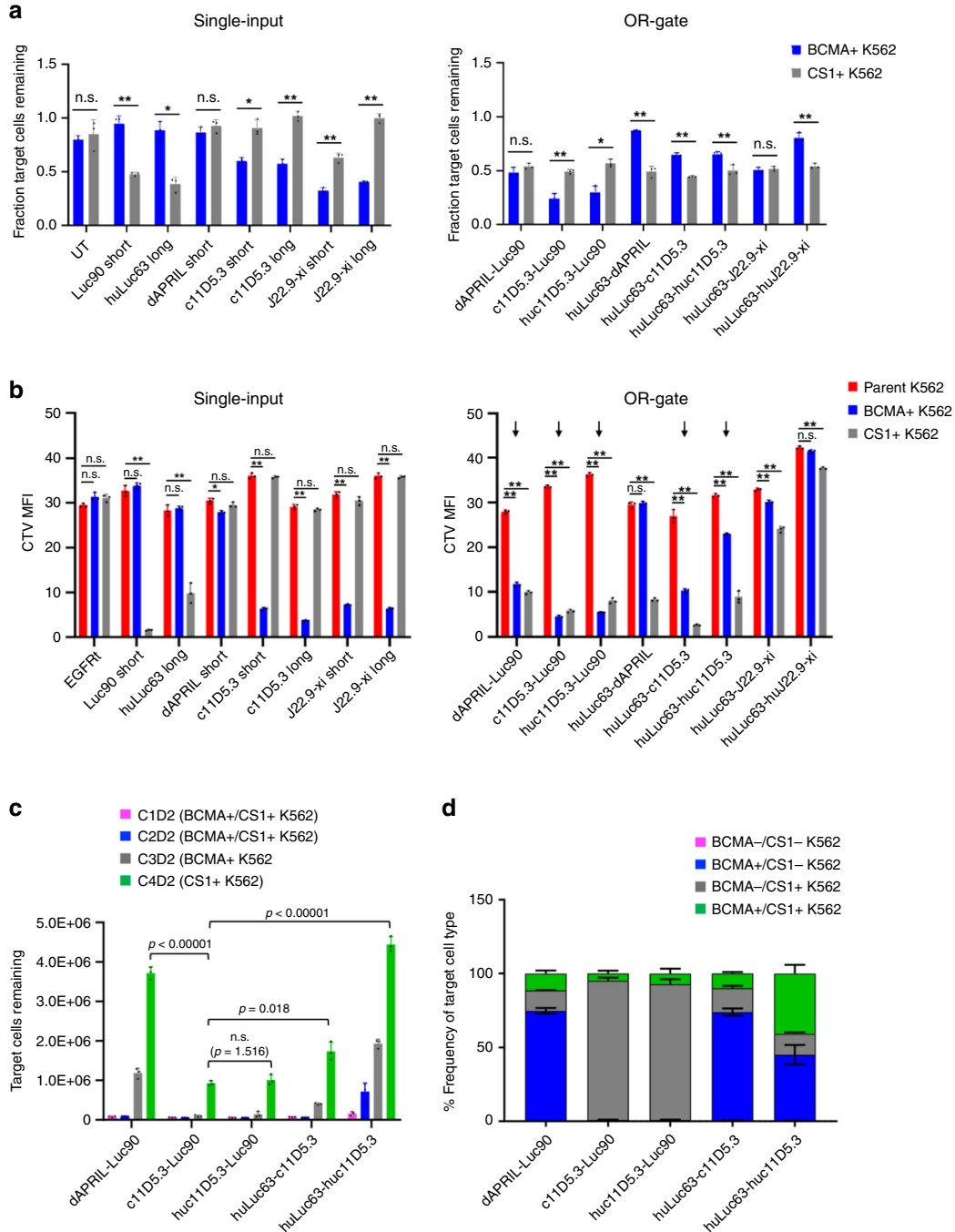

**Fig. 2 OR-gate CAR panel exhibits range of efficacy against BCMA+ and CS1+ targets. a** Cell-lysis activity of single-input and bispecific CD8+ CAR-T cells against K562 targets engineered to express either BCMA or CS1. Cells were seeded at an effector-to-target (E:T) ratio of 2:1, where effector-cell seeding was based on CAR+ T-cell count. The fraction of viable K562 cells left after a 20-h coincubation was quantified by fluorescence imaging of target cells using IncuCyte. All bispecific CARs in this panel contained a short extracellular spacer. **b** Proliferation of single-input and bispecific BCMA/CS1 CD8 + CAR-T cells upon antigen stimulation. CAR-T cells were stained with CellTrace Violet (CTV) dye. CTV median fluorescence intensity (MFI) was quantified by flow cytometry after a 5-day coincubation with parental (BCMA−/CS1−), BCMA+, or CS1+ K562 target cells at a 2:1 E:T ratio. CARs containing huLuc63 paired with dAPRIL, J22.9-xi, and huJ22.9-xi were subsequently eliminated from the panel based on poor cytotoxicity and/or T-cell proliferation. **c** Cytotoxicity of reduced bispecific CAR-T-cell panel upon repeated antigen challenge. CD8+ CAR-T cells were coincubated with K562 target cells at a 1:1 E:T ratio and rechallenged every 2 days with fresh target cells. Viable target-cell count was quantified by flow cytometry 2 days after each target-cell addition. 'C#' denotes the challenge number and 'D#' denotes the number of days post challenge. **d** Characterization of the remaining K562 target-cell populations after four challenges from **c** reveals differing antigen preference among the panel of bispecific CARs. Values shown are the means of technical triplicate samples, with error bars indicating +1 standard deviation (SD). P-values were calculated by unpaired two-tailed Student's t-test; n.s. not statistically significant (p > 0.05); *p < 0.05; **p < 0.01, with Bonferroni correction for multiple comparisons applied. P-values in **c** were calculated for the final time point for each construct, relative to the top-performing CAR, c11D5.3-Luc90. Source data are provided as a Source Data File. P-values in **a** and **b** can be referenced in the Source Data File.

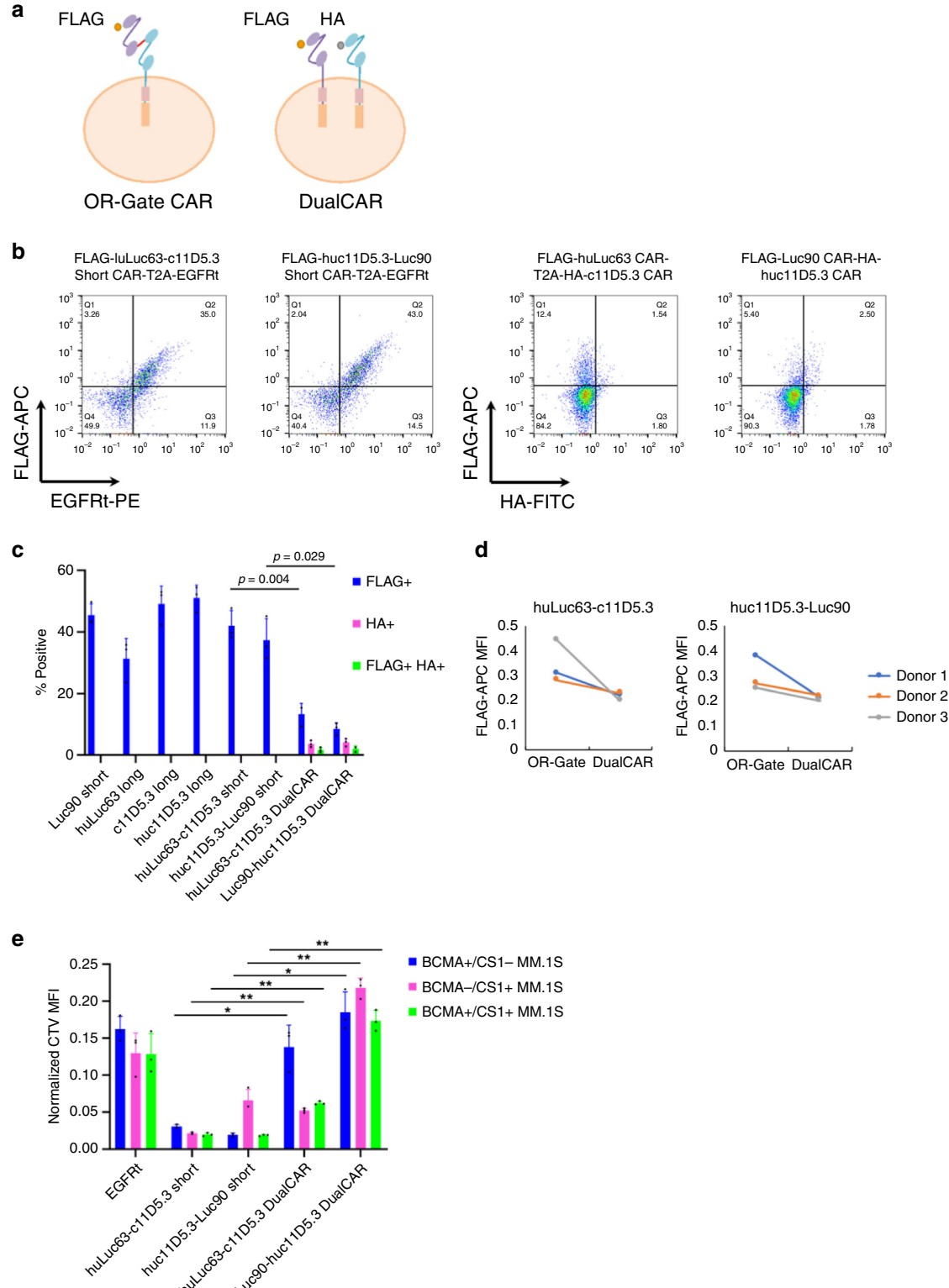

to be the least effective among the three strategies noted above both in vitro and in vivo[11]. As such, we chose to focus on comparing the OR-gate vs. DualCAR approaches (Fig. 3a).

We reasoned that the single-chain OR-gate CARs should be more efficiently integrated and expressed due to their compact size, thus yielding a more functional CAR-T-cell product compared to the DualCAR approach. This hypothesis was experimentally verified when we compared T cells expressing either an OR-gate CAR or the corresponding pairs of single-input

CARs encoded in bicistronic cassettes connected by a 2A sequence (Supplementary Fig. 4b). Flow-cytometry analysis revealed that the DualCAR-T cells had substantially lower CAR surface expression, in terms of % CAR+ as well as median fluorescence intensity (MFI) for CAR expression, compared to OR-gate CAR-T cells (Fig. 3b–d). Furthermore, DualCAR-T cells exhibited significantly weaker T-cell proliferation upon antigen stimulation (Fig. 3e) compared to the corresponding OR-gate CAR-T cells, even when the assay setup was normalized by CAR+

**Fig. 3 OR-gate CAR-T-cells outperform T cells co-expressing two separate CARs. a** Single-input and bispecific CARs were tagged with an N-terminal FLAG tag. In DualCAR constructs, the CS1 CAR was N-terminally tagged with a FLAG tag while the BCMA CAR was N-terminally tagged with a HA tag. **b** CAR surface expression levels were quantified by surface antibody staining of FLAG and HA tags followed by flow cytometry. Each single-input and single-chain bispecific CAR was tagged with an N-terminal FLAG. The first CAR in each DualCAR construct was tagged with the FLAG while the second CAR was tagged with HA. See Supplementary Fig. 4b for construct schematics. **c** Transduction efficiency as measured by % CAR+ (FLAG+, HA+, or FLAG+HA+) Data shown represent the mean value from three different donors, with error bars indicating +1 SD. **d** Median fluorescence intensity (MFI) of FLAG antibody staining for T cells expressing either OR-gate or DualCAR constructs. **e** NM CAR-T-cell proliferation following a 5-day coincubation with MM.1 S target cells. Values shown are the means of technical triplicate samples from the same donor, with error bars indicating +1 SD. Data shown are representative of results from two independent experiments performed with cells from two different healthy donors. P-values for all panels were calculated by unpaired two-tailed Student's t-test; n.s. not statistically significant ($p > 0.05$); *$p < 0.05$; **$p < 0.01$, with Bonferroni correction for multiple comparisons applied. P-values in **c** were calculated for the OR-gate CAR constructs, relative to their corresponding DualCAR constructs. Source data are provided as a Source Data File. P-values in **e** can be referenced in the Source Data File.

T-cell count. Lysis assays normalized by CAR+ T-cell count were also attempted. However, the low % CAR+ among the DualCAR samples (0.97–2.56% DualCAR+ across three different donors, Fig. 3c) necessitated the inclusion of a large number of untransduced T cells to normalize for CAR+ T-cell input across samples. At that cell density, nonspecific killing dominated the system, and no difference in killing could be observed between the mock-transduced control and CAR-expressing samples (Supplementary Fig. 6). Taken together, these observations highlight the importance of high transduction efficiency afforded by the compact genetic footprint of the single-chain bispecific CAR architecture, consistent with previously published results[11].

**Absence of fratricide by OR-gate CAR-T cells.** In addition to being highly expressed on myeloma cells, CS1 is also expressed at low yet detectable levels on other hematopoietic cells, including CD8+ T cells (Supplementary Figs. 7a and 8a)[17,25]. Therefore, the optimal CAR-T cells for MM therapy should be able to distinguish between high and low levels of CS1 expression and robustly target MM tumors while sparing healthy cells. In principle, the functional assays performed in our OR-gate CAR development process should select against constructs that would trigger T-cell fratricide, since such behavior would reduce the overall viability and thus potency of the cell product. To specifically evaluate our top two OR-gate CAR candidates for the propensity for fratricide, OR-gate CAR-T cells were coincubated with donor-matched, untransduced, CellTrace Violet (CTV)-labeled CD8+ T cells, whose survival was quantified after a 24-h coincubation. Results showed no significant difference in either the killing of bystander CD8+ T cells or ex vivo culture expansion by OR-gate CAR-T cells in comparison to single-input BCMA CAR-T cells (c11D5.3 Long) or mock-transduced T cells (Supplementary Figs. 7b, c and 8b). Interestingly, OR-gate CAR-T cells showed superior performance compared to single-input CS1 CAR-T cells (Luc90 Short and huLuc63 Long) upon repeated antigen challenge (Supplementary Fig. 7d, e). A likely explanation is that the OR-gate CAR-T cells have slightly weaker reaction to CS1+ target cells compared to the single-input CS1 CAR-T cells, striking a balance that enables robust tumor killing without inducing premature functional exhaustion of the T cells.

**Superior antitumor function by naïve/memory T cells.** Functional testing of the OR-gate CARs designed in this study were initially performed using bulk-sorted CD8+ T cells. However, it had been shown that administering a mixture of CD8+ and CD4+ T cells could improve performance over CD8+ T cells alone[26], and that T cells exhibiting a memory phenotype could improve CAR-T-cell persistence and function in vivo[26–29]. Therefore, we next compared CD8-derived CAR-T cells against CAR-T cells derived from a naïve/memory (NM) starting

population, which is obtained by subjecting peripheral blood mononuclear cells (PBMCs) to CD14 and CD25 depletion to remove monocytes and regulatory T cells, followed by CD62L enrichment to select for naïve and memory T cells.

Across all CAR constructs, NM-derived CAR-T cells demonstrated substantially higher cytokine production, more sustained target-cell killing upon repeated challenge, and greater T-cell proliferation compared to CD8-derived CAR-T cells (Supplementary Fig. 9). To examine whether the functional improvement seen in NM-derived cells is simply due to the added presence of CD4+ cells, or due to biology that is specific to naïve and memory T cells, we further compared NM-derived CAR-T cells with CAR-T cells generated from a CD3-sorted population (i.e., total T cells) in subsequent in vivo studies. NSG mice bearing established wildtype BCMA+/CS1+ MM.1 S xenografts were treated with $0.5 \times 10^6$ NM-derived, CD3-derived, or CD8-derived c11D5.3-Luc90 OR-gate CAR-T cells. NM-derived OR-gate CAR-T cells achieved initial tumor clearance more rapidly compared to the other two cell types (Supplementary Fig. 10a) and showed the highest overall median survival (Supplementary Fig. 10b), with one mouse exhibiting complete tumor clearance following T-cell redose on day 21 (Supplementary Fig. 10a). Analysis of tumor cells recovered at the time of sacrifice indicated that the cells retained antigen expression (Supplementary Fig. 10c), thus the failure to eradicate tumors was not a result of spontaneous antigen escape and likely attributable to the tumor model's aggressiveness and the low T-cell dose administered. Based on these results, NM-derived T cells were chosen for subsequent studies.

**Bispecific BCMA/CS1 CARs outpace single-input CARs in vivo.** To evaluate the ability of OR-gate CAR-T cells to prevent antigen escape by heterogeneous tumors in vivo, NSG mice were engrafted with a mixed population of three firefly luciferase-expressing MM.1 S cell lines, containing a 1:1:1 ratio of BCMA+/CS1+, BCMA+/CS1−, and BCMA−/CS1+ MM.1 S cells (Supplementary Fig. 11). Tumor-bearing mice were treated with single-input or OR-gate CAR-T cells on days 5 and 13 post tumor injection. The tumor and T-cell dosages were chosen to result in an aggressive and recalcitrant tumor model, so as to allow "stress-testing" of the different CAR-T cells and identification of the truly superior CAR design. Bioluminescence imaging revealed huLuc63-c11D5.3 Short CAR-T cells as the clear leader in antitumor activity, yielding near-complete tumor clearance by day 12 (Fig. 4a). Notably, animals treated with single-input BCMA or CS1 CAR-T cells fared no better than those treated with mock-transduced (EGFRt-expressing) T cells (Fig. 4b, Supplementary Fig. 12a), whose antitumor efficacy was presumed to originate from allogeneic effects[30]. In contrast, animals in the huLuc63-c11D5.3 Short CAR-T-cell-treated groups showed significantly longer median survival, and one animal achieved complete and

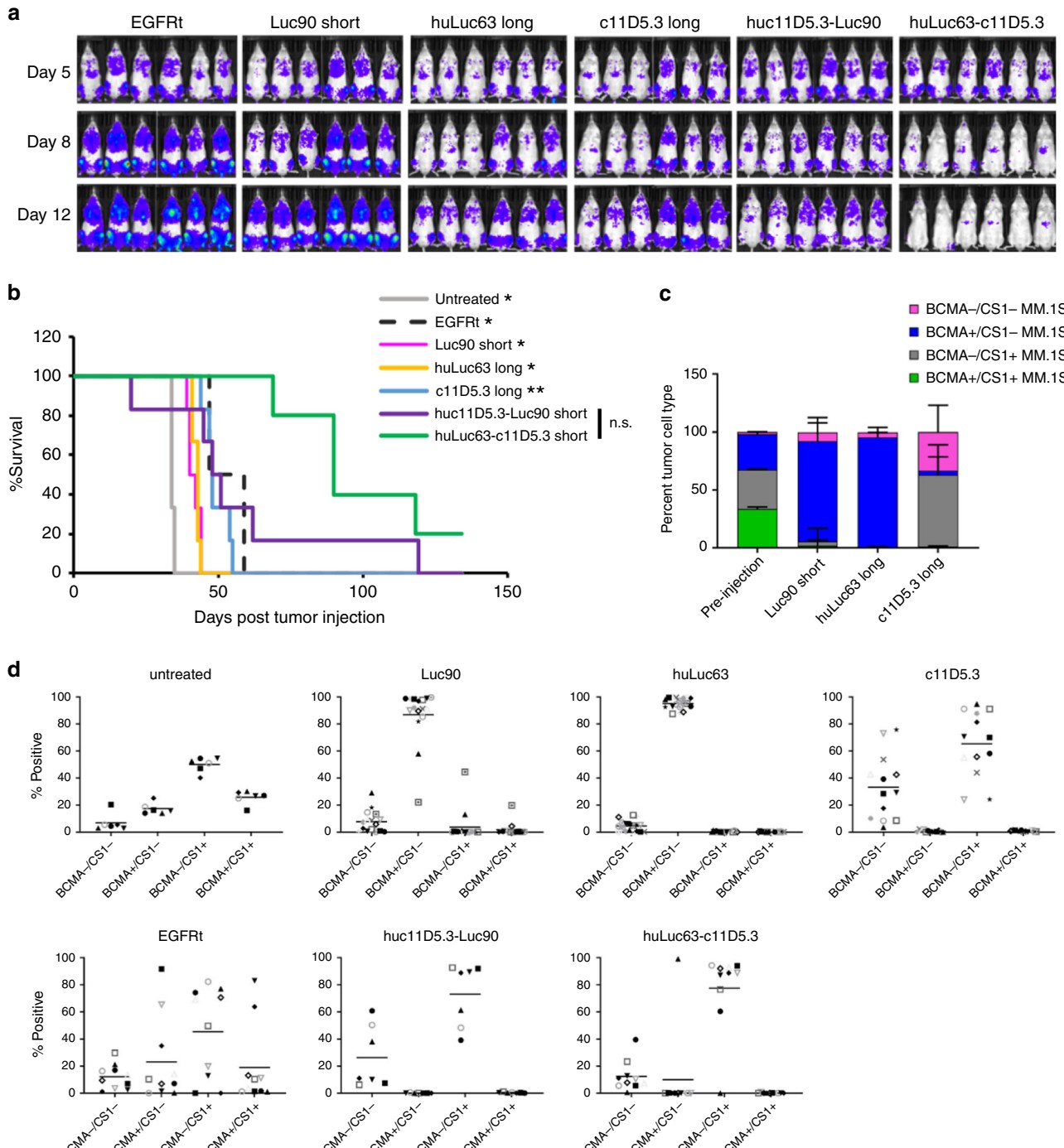

**Fig. 4 BCMA/CS1 OR-gate CAR-T cells prevent antigen escape in vivo. a** Evaluation of single-input and bispecific CAR-T cells in vivo. Mice were engrafted with a mixture of $1.5 \times 10^6$ MM.1 S cells containing a 1:1:1 ratio of BCMA$^+$/CS1$^-$, BCMA$^-$/CS1$^+$, and BCMA$^+$/CS1$^+$ cells. Tumor-bearing animals were treated with $1.5 \times 10^6$ EGFRt- or CAR-expressing T cells on day 5 (5 days after tumor injection) and day 13. Six mice were included in each initial treatment group but only five mice in the huLuc63-c11D5.3 group were redosed due to limited T-cell availability. Tumor progression was monitored by bioluminescence imaging. **b** Survival of mice shown in **a**. Statistical difference (depicted) between survival of huLuc63-c11D5.3-treatment group compared with other treatment groups was determined using log-rank analysis, applying a Chi-square distribution with one degree of freedom; n.s. not statistically significant ($p > 0.05$); *$p < 0.05$; **$p < 0.01$. P-values for the different treatment groups are as follows: Untreated, $p = 0.024$; EGFRt, $p = 0.024$; Luc90, $p = 0.011$; huLuc63, $p = 0.015$; c11D5.3, $p = 0.008$; huc11D5.3-Luc90, $p = 0.114$. **c** BCMA/CS1 antigen expression on tumors harvested from mice treated with CS1 single-input Luc90 Short, huLuc63 Long, or BCMA single-input c11D5.3 Long CAR-T cells. Values shown are the means of $n = 4$ technical replicates for cells stained prior to injection. Thirteen to 18 tumor samples were collected from different locations in the mice (six mice per test group) at the time of sacrifice, with error bars indicating +1 standard deviation (SD), and each sample corresponding to 100 or more events as detected by flow cytometry. **d** Antigen expression pattern on tumor cells recovered at the time of animal sacrifice. Each data point **d** corresponds to an individual tumor sample recovered that included more than 100 tumor cells as detected by flow cytometry; each mouse generally contained multiple tumors at the time of sacrifice. Source data are provided as a Source Data File.

durable tumor clearance through the 134-day study (Fig. 4b and Supplementary Fig. 12b).

MM.1 S cells recovered from tumor-bearing animals at the time of animal sacrifice revealed an intriguing pattern of antigen expression. Although all MM.1 S cells expressed at least one antigen at the time of tumor injection (Supplementary Fig. 11), a substantial fraction of tumor cells recovered from animals treated with single-input BCMA CAR-T cells were BCMA$^-$/CS1$^-$ (mean 33%, range 4–76%), suggesting some MM.1 S cells had spontaneously lost BCMA expression under selective pressure from BCMA CAR-T cells (c11D5.3 Long, Fig. 4c). This double-negative tumor population was not observed in significant numbers in untreated animals or animals treated with single-input CS1 CAR-T cells (Luc90 Short and huLuc63 Long; Fig. 4c, Supplementary Fig. 13a), underscoring BCMA's particular vulnerability to antigen escape when treated with single-input BCMA CAR-T cells.

We further observed that tumors remaining in OR-gate CAR-treated animals were mostly BCMA$^-$/CS1$^+$, with a minor population of double-negative tumors (Fig. 4d). Two predominantly BCMA$^-$/CS1$^+$ tumor samples recovered from two different animals in the huLuc63-c11D5.3 Short CAR-T-cell-treated group were analyzed by amplicon sequencing, and results indicated that both tumors originated from cells that had been engineered by CRISPR/Cas9 editing to be BCMA$^-$ prior to in vivo engraftment (Supplementary Fig. 14 and Supplementary Data File 1). Three non-mutually exclusive possibilities could explain the particularly strong persistence of BCMA$^-$/CS1$^+$ tumor cells in animals treated with OR-gate CAR-T cells: (i) the OR-gate CAR-T cells were less effective against CS1 than against BCMA, (ii) the residual tumor cells have become unrecognizable or resistant to CAR-T cells, or (iii) the BCMA$^-$/CS1$^+$ tumor line has an inherent growth advantage over the wildtype and BCMA$^+$/CS1$^-$ MM.1 S lines. We empirically evaluated each possibility in turn.

First, time-lapse imaging analysis (IncuCyte) was performed to quantify the kinetics of tumor-cell killing by OR-gate CAR-T cells in vitro. Results indicated huLuc63-c11D5.3 Short CAR-T cells killed CS1$^+$ tumor cells more rapidly than BCMA$^+$ targets (Supplementary Fig. 15), which is consistent with previous observations (Fig. 2d) and argues against the hypothesis that OR-gate CAR-T cells were less effective against the CS1 antigen.

Second, tumor cells recovered from two different animals in the huLuc63-c11D5.3 Short CAR-T-cell-treated group (same as those analyzed in Supplementary Fig. 14) were expanded ex vivo and rechallenged by huLuc63-c11D5.3 Short CAR-T cells. Both tumor samples maintained relatively stable antigen expression profiles during ex vivo cell expansion (Supplementary Fig. 16), and both were efficiently eliminated by the CAR-T cells, indicating that the tumor cells remained recognizable and vulnerable to CAR-T cells (Supplementary Fig. 17a, b).

Third, wildtype, BCMA$^+$/CS1$^-$, and BCMA$^-$/CS1$^+$ MM.1 S cells were co-cultured at 1:1:1 ratio in vitro, and no difference in their relative growth rate was observed over a 5-week period (Supplementary Fig. 17c). However, tumors recovered from animals that were either untreated or treated with mock-transduced (EGFRt) T cells also showed an enrichment of BCMA$^-$/CS1$^+$ cell content despite the lack of selective pressure against either antigen (Fig. 4d), suggesting that the BCMA$^-$/CS1$^+$ cell-line may have a growth advantage in the in vivo milieu that is not evident in cell culture. Amplicon sequencing results from tumor samples indicate that the vast majority of cells in each tumor arose from a single clone of MM.1 S, but two different clones gave rise to the two tumors that were sequenced. This was evidenced by the fact that within each tumor,

nearly all cells (99.5–99.7%) contained the same BCMA mutation, but the two tumors contained two different BCMA mutations within the CRISPR-edited region (Supplementary Data File 1). Therefore, an intriguing possibility is that BCMA$^-$/CS1$^+$ cells may have greater capacity to undergo in vivo clonal expansion compared to WT or BCMA$^+$/CS1$^-$ cells.

**Virally integrated CARs outperform gene-edited counterparts.** Having identified the lead OR-gate CAR candidate through in vivo testing, we next evaluated whether alternative manufacturing processes may further bolster T-cell function. It had been reported that CAR-T cells with the CAR integrated into the T-cell receptor α constant (TRAC) locus via homology-directed repair (HDR) exhibit longer T-cell persistence and less exhaustion upon antigen stimulation in vivo compared to retrovirally transduced CAR-T cells[31]. It was hypothesized that the endogenous gene-expression regulation machinery of the TRAC locus enabled dynamic regulation of CAR expression, leading to superior functional output[31]. We thus integrated a FLAG-tagged huLuc63-c11D5.3 OR-gate CAR into the TRAC locus (Supplementary Fig. 18a) and verified TRAC knockout and CAR knock-in by surface antibody staining for TCR α/β chains and the FLAG-tag, respectively (Supplementary Figs. 18b, 19).

Consistent with previous report, CAR expression from the TRAC locus is lower than that detected in virally transduced cells (Supplementary Fig. 18c)[31]. TRAC-knockout T cells showed comparable viability to that of lentivirally transduced cells, indicating CRISPR/Cas9-mediated editing through RNP nucleofection did not compromise cell viability (Supplementary Fig. 18d). However, contrary to expectations, TRAC-knockout T cells that were HDR-modified to express OR-gate CARs showed poor viability and inferior cytotoxicity upon repeated antigen challenge compared to lentivirally transduced OR-gate CAR-T cells (Supplementary Fig. 18d, e). Furthermore, HDR-modified cells showed weaker antigen-stimulated T-cell proliferation (Supplementary Fig. 18f), as well as higher and more durable exhaustion-marker expression (Supplementary Fig. 18g), compared to lentivirally transduced cells. These results suggest that whether site-specific CAR integration into the TRAC locus is beneficial to T-cell function may depend on the specific CAR construct involved. Based on these findings, lentiviral transduction was retained as the preferred method for CAR-T-cell generation.

**Bispecific CAR-T-cell and anti-PD-1 combination therapy.** Tissue recovered at the time of animal sacrifice in the in vivo study shown in Fig. 4 revealed the presence of CAR-T cells, but they were generally present at low frequency and with high PD-1 expression (Supplementary Figs. 13b, 20). This observation suggested combination therapy with checkpoint inhibitors may improve treatment efficacy. Indeed, we found that coadministration of anti-PD-1 antibody and the huLuc63-c11D5.3 OR-gate CAR-T cells led to more effective tumor control compared to OR-gate CAR-T cells alone at early time points. By day 48 post T-cell injection (day 56 post tumor injection), 5/6 animals treated with OR-gate CAR-T cells plus anti-PD-1 exhibited complete tumor clearance, compared to 1/6 animal in the group treated with CAR-T cells alone (Fig. 5a). The beneficial effect of checkpoint inhibition was dependent on the presence of CAR-T cells, as anti-PD-1 therapy alone or in combination with mock-transduced T cells did not confer antitumor capability. In fact, the addition of anti-PD-1 antibodies appeared to reduce the allogeneic effect exerted by mock-transduced T cells on engrafted tumors (Fig. 5a).

Although CAR-T-cell treatment alone failed to control initial tumor progression in most animals, two mice that had developed

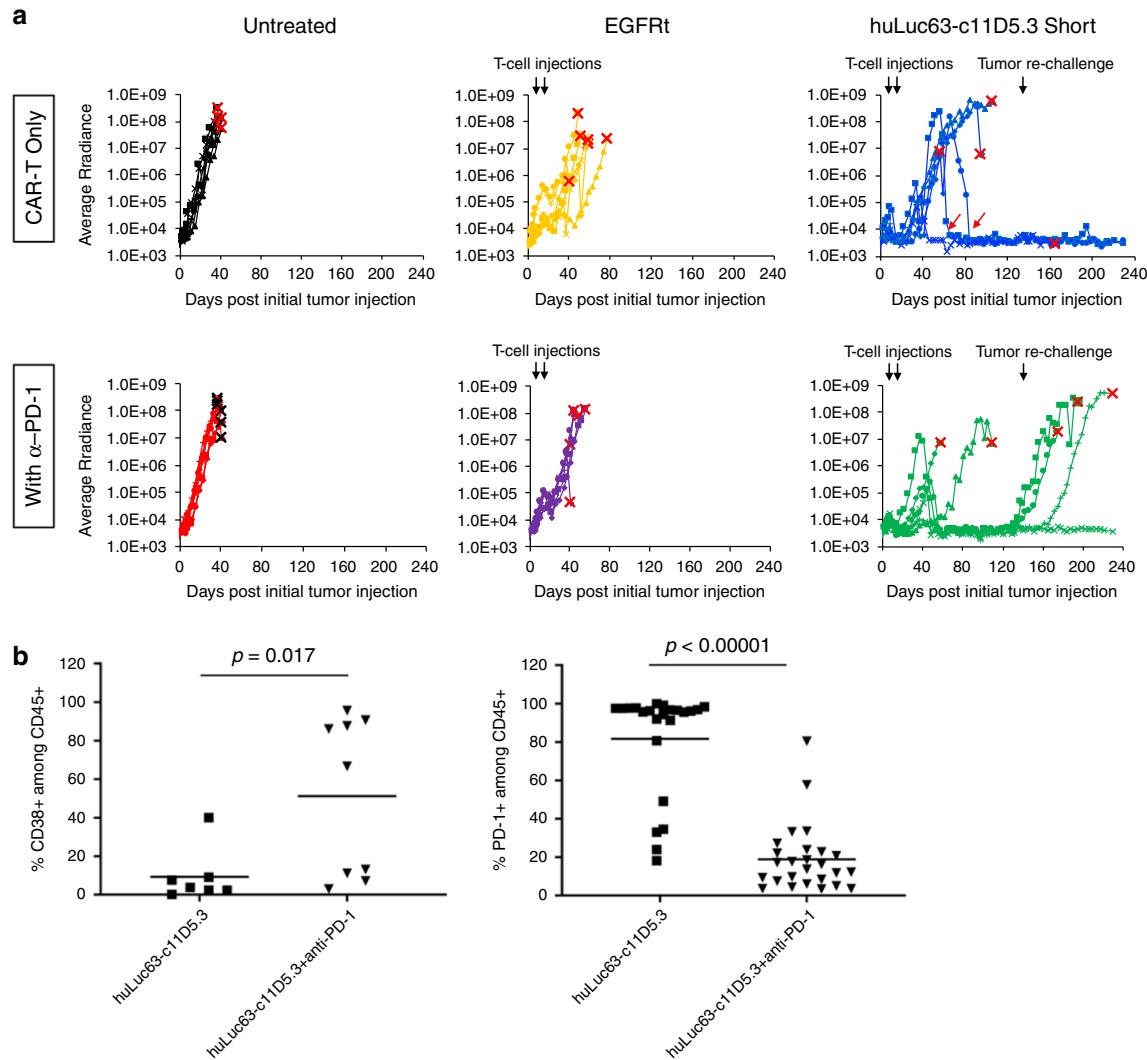

**Fig. 5 Combination therapy with anti-PD-1 increases initial antitumor efficacy but not durability of response in vivo. a** Mice were engrafted with $1.5 \times 10^6$ WT MM.1 S cells. Tumor-bearing animals were treated with $1.5 \times 10^6$ EGFRt- or CAR-expressing T cells on day 8 (8 days after tumor injection) and day 16. Tumor progression was monitored by bioluminescence imaging and shown for individual animals in each test group ($n = 6$). On day 133, animals that had been tumor-free for at least 7 weeks (3 each in the huLuc63-c11D5.3 Short groups with and without anti-PD-1) were rechallenged with $1.5 \times 10^6$ WT MM.1 S cells. Black arrows indicate time of cell injections; red arrows indicate instances of CAR-T cells eradicating palpable tumor nodules. The endpoint for each animal is marked with an "X." **b** CD38 and PD-1 expression on T cells persisting in mice at time of animal sacrifice. Each data point in **b** corresponds to an individual tumor mass, tissue (e.g., brain and spleen), or cardiac blood sample recovered. For PD-1 expression: 23 samples were collected from six mice in the group not treated with anti-PD-1, and 26 samples were collected from six mice in the group treated with anti-PD-1. For CD38 expression: seven samples were collected from two mice in the group not treated with anti-PD-1, and nine samples were collected from three mice in the group treated with anti-PD-1. Only samples that contained at least 10 human $CD45^+$ cells as detected by flow cytometry were included for analysis. *P*-values were calculated by unpaired two-tailed Student's *t*-test. Source data are provided as a Source Data File.

palpable solid tumors became tumor-free 50 and 68 days after the second and final T-cell infusion, respectively (red arrows in Fig. 5a; Supplementary Fig. 21). This result demonstrates BCMA/CS1 OR-gate CAR-T cells' ability to eradicate established solid-tumor masses even after a prolonged period in vivo. Notably, all animals that achieved complete tumor clearance in the "CAR-T only" treatment group remained tumor-free over the entire study duration, even after tumor rechallenge performed on day 133 (117 days after the second and last T-cell infusion, Fig. 5a). One of the animals in this group was sacrificed on day 165 of the study due to symptoms consistent with graft versus host disease, in the complete absence of tumor signal (Supplementary Fig. 22).

In a surprise finding, animals treated with OR-gate CAR-T cells plus anti-PD-1 showed a higher rate of relapse at later time points compared to those treated with CAR-T cells alone.

Specifically, two out of five animals that achieved initial tumor clearance experienced relapse before the time of tumor rechallenge, and another two failed to prevent tumor outgrowth after rechallenge (Fig. 5a). Verma et al. recently reported that PD-1 blockade in suboptimally primed T cells can induce a dysfunctional $PD-1^+/CD38^+$ population, resulting in decreased antitumor efficacy[32]. In our study, animals were given twice-weekly injections of anti-PD-1 even after they have achieved tumor clearance, thus the CAR-T cells may have become dysfunctional due to continual exposure to anti-PD-1 in the absence of antigen stimulation. Consistent with this hypothesis, we found a higher frequency of $CD38^+$ cells among T cells recovered from animals treated with anti-PD-1 (Fig. 5b). T cells harvested from this group of animals also showed significantly lower PD-1 staining. However, this may be due to the fact that the PD-1 molecules

are already masked by the anti-PD-1 antibodies administered to the animals. (Clone EH12.2H7 was used for mouse treatment; clone PD1.3.1.3 was used for flow-cytometry analysis after T-cell recovery. Zelba et al. reported that both EH12.2H7 and PD1.3.1.3 compete with pembrolizumab for antigen binding, suggesting that the three monoclonal antibodies all bind to overlapping epitopes[33]). Taken together, these results suggest that administration of anti-PD-1 could accelerate tumor clearance at early time points and increase the initial rate of response, but prolonged anti-PD-1 administration may not provide added benefits to BCMA/CS1 CAR-T-cell therapy. Large-scale animal studies that track animal response over an extended time period would be advisable before ascertaining the benefit of combination immunotherapies.

## Discussion

Following the success of CD19 CAR-T-cell therapy for B-cell leukemia and lymphoma, the BCMA CAR is a leading candidate to receive the next FDA approval for adoptive T-cell therapy for cancer. However, outcomes from recent clinical trials indicate that BCMA-targeted CAR-T-cell therapy is vulnerable to antigen escape[4,5,7]. To develop a more effective CAR for MM treatment, we engineered single-chain bispecific (OR-gate) CARs that efficiently target not only BCMA but also CS1. Via high-throughput CAR construction and screening as well as systematic optimization of the cell-manufacturing process, we generated BCMA/CS1 OR-gate CAR-T cells that can robustly eliminate heterogeneous MM cells in vitro and in vivo.

CS1 is expressed in more than 90% of patient MM samples and not expressed on nonhematological and essential tissues such as the stomach, lung, kidney, brain, and heart[17,18,25]. A previous study has shown that CS1 is more commonly expressed than BCMA on patient-derived MM samples[34]. As such, CS1 is an ideal target to be paired with the more heterogeneously expressed but clinically validated BCMA for MM treatment. However, CS1 is expressed on natural killer (NK) cells, natural killer T (NKT) cells, CD8$^+$ T cells, activated monocytes, and dendritic cells, albeit at much lower levels than on plasma cells[17,25]. CS1 expression on non-cancerous hematological cell types raises the question of potential off-tumor toxicities, but this concern is partially allayed by the fact that the FDA-approved anti-CS1 antibody elotuzumab (i.e., huLuc63) showed no evidence of increased autoimmunity or immune-related adverse events in the clinic[35–37]. Consistent with these clinical observations, we noted that CS1-specific CAR-T cells showed slightly but not statistically significantly higher lytic activity against bystander CD8$^+$ T cells, and they showed no defects in ex vivo expansion (Supplementary Fig. 7b, c).

In principle, given CS1's nearly uniform expression on MM cells, a CS1 single-input CAR-T-cell therapy may be adequate. However, we observed that single-input CS1 CAR-T cells show signs of functional defect upon repeated antigen challenge in vitro (Supplementary Fig. 7d, e), and they are less potent in vivo compared to single-input BCMA CAR-T cells (Fig. 4a, b). A previous study had also demonstrated that single-input CS1 CAR-T cells require the assistance of lenalidomide administration to achieve optimal therapeutic outcome[19]. Taken together, these results suggest that CS1-targeting as a single therapy may not be maximally efficacious. By combining both BCMA and CS1 in a single-chain bispecific CAR design, we take advantage of both the uniform expression of CS1 and the strong antitumor output elicited by BCMA targeting to achieve more effective tumor control.

To arrive at a bispecific CAR with robust activity against both antigens, we systematically compared 13 CAR variants that combinatorially sampled three different BCMA-binding moieties and two different CS1-specific scFvs. Importantly, we made decisions on which specific CAR variants to build and test based on available structural information for the CS1-binding epitopes, rather than indiscriminately testing all possible combinations or restricting our designs based on prior examples of CS1- or BCMA-targeting CARs. This approach allowed us to rapidly identify highly functional variants and revealed several unexpected findings, such as the relatively weak performance of dAPRIL-containing CARs against BCMA.

In addition to the single-chain bispecific "OR-gate" CAR described here, other strategies can be taken to demonstrate bispecific targeting, including DualCAR (co-expressing two full-length receptors on one cell) and CARpool (combining two single-input CAR-T-cell products). Compared to the OR-gate CAR, the DualCAR requires a much larger genetic payload, which leads to poor transduction efficiency and reduced antigen-stimulated proliferation (Fig. 3). These observations are consistent with previous reports of functional superiority achieved with single-chain bispecific CARs relative to DualCAR constructs, and the fact that compact genetic footprints can greatly facilitate viral integration and thus cell-product manufacturing[11,38–40]. A CARpool strategy could avoid the issue of poor transduction efficiency, but it requires manufacturing two clinical products and, perhaps more importantly, compels the two engineered T-cell populations to compete for the limited nutrients and homeostatic cytokines available in circulation. Indeed, a previous study reported that, among the three bispecific approaches discussed above (OR-gate, DualCAR, or CARpool), the CARpool strategy is the least effective while the OR-gate CAR approach is the most effective both in vitro and in vivo[11].

In this study, we further demonstrated that the starting T-cell population can affect the functionality of final cell products (Supplementary Figs. 9 and 10), highlighting the importance of cell-manufacturing parameters to eventual therapeutic efficacy. Furthermore, we found that lentivirally transduced CAR-T cells exhibit greater antitumor activity compared to CAR-T cells generated through CRISPR/Cas9-mediated editing (Supplementary Fig. 18). This finding was unexpected given the compelling data from a previous study demonstrating functional superiority of T cells that had undergone site-specific integration of the CAR transgene into the TRAC locus[31]. Our findings suggest that further exploration may be warranted to determine whether the benefit of site-specific CAR transgene integration is limited to some CARs and, if so, whether the difference is determined by the antigen specificity of the CAR or more generalizable properties such as the size of the CAR construct (and thus the size of the HDR template).

Our in vivo study demonstrated that rationally optimized BCMA/CS1 OR-gate CAR-T cells are uniquely capable of controlling heterogeneous MM that proves resistant to single-input BCMA or CS1 CAR-T-cell therapy (Fig. 4). Our in vivo data revealed intriguing dynamics of MM evolution under selective pressure, with results indicating that BCMA may be particularly susceptible to antigen escape under selective pressure from single-input BCMA CAR-T-cell therapy (Fig. 4d), and underscores the utility of dual-antigen targeting for MM.

Multiple ongoing clinical trials are investigating the effect of combining CAR-T-cell therapy with checkpoint inhibitors (e.g., NCT04003649, NCT00586391, NCT03726515), reflecting interest in the field to explore synergy between these two potent therapeutic paradigms. We demonstrated that the coadministration of anti-PD-1 antibody with OR-gate CAR-T cells can accelerate the rate of initial tumor eradication in animals that had been engrafted with highly aggressive MM xenografts, but continued anti-PD-1 administration may not provide added benefits in the long run (Fig. 5). A recent study reported that PD-1 blockade

actually induces dysfunction in suboptimally primed T cells[32]. In our study, anti-PD-1 treatment showed clear beneficial effects at early time points, when tumor burden was still present. After day 60, the vast majority of OR-gate CAR-T-cell-treated animals had achieved tumor-free status and thus ceased to provide antigen stimulation to T cells. It is plausible that continued administration of anti-PD-1 induced T-cell dysfunction during the >2-month period between initial tumor clearance and tumor rechallenge on day 133, causing antibody-treated animals to eventually fail to resist the second tumor challenge. Taken together, our findings suggest that combining CAR-T-cell and checkpoint inhibitor therapies requires judicious examination of the timing and duration of the combination therapy, and an even larger-scale in vivo study would be warranted to clearly understand the effect of long-term administration of anti-PD-1 antibody after CAR-T-cell therapy.

This work presents a rational approach for the engineering of BCMA/CS1 OR-gate CAR-T cells that can effectively target MM tumors and substantially reduce the probability of tumor antigen escape. The small genetic footprint of OR-gate CAR constructs facilitate the clinical manufacturing of T-cell products, and OR-gate CAR-T cells' functional superiority over DualCAR-T cells provide a compelling advantage for clinical translation. Finally, the vertically integrated optimization process outlined in this study can be applied towards the engineering of novel CARs to expand the applications of adoptive T-cell therapy to additional cancer types currently lacking effective treatment options.

## Methods

**Plasmid construction.** Single-chain bispecific BCMA-OR-CS1 CARs were constructed by isothermal assembly[41] of DNA fragments encoding the following components. BCMA-specific single-chain variable fragments (scFvs) were derived from either the c11D5.3[42,43] or the J.22-xi[44] monoclonal antibody (mAb), and dAPRIL[45] was also evaluated as an alternative BCMA-binding domain. CS1-specific scFvs were derived from Luc90 or huLuc63 mAb[46,47]. Each CAR also contained an IgG4-based extracellular spacer, the CD28 transmembrane domain, and the cytoplasmic domains of 4-1BB and CD3ζ. Amino-acid sequences of all CAR components are shown in Supplementary Table 1. All CARs were fused to a truncated epidermal growth factor receptor (EGFRt) via a T2A peptide to facilitate antibody staining of CAR-expressing cells[48]. An N-terminal FLAG or HA tag was also added to each CAR to enable quantification of CAR surface expression.

**Cell-line generation and maintenance.** Parental K562 cells were a gift from Dr. Michael C. Jensen (Seattle Children's Research Institute) received in 2011. Antigen-expressing K562 cells were generated by retrovirally transducing parental K562 with BCMA- and/or CS1-encoding constructs. MM.1 S cells (ATCC) were lentivirally transduced to express the EGFP–firefly luciferase (ffLuc) fusion gene, and EGFP+ cells were enriched by fluorescence-activated cell sorting (FACS) to >98% purity. BCMA− or CS1− MM.1 S cells were generated by CRISPR/Cas9-mediated gene knockout. MM.1 S cells ($5 \times 10^6$) were nucleofected with ribonucleoprotein (RNP), consisting of chemically synthesized gRNA (Synthego) targeting BCMA or CS1 complexed to purified Cas9 protein, using Ingenio Electroporation Solution (Mirus Bio) and the Amaxa Nucleofector 2B Device (Lonza) following manufacturers' protocols. Four days after nucleofection, cells were surface-stained with BCMA-PE and CS1-APC antibodies (BioLegend) to verify antigen knockout. The cells were subsequently bulk-sorted for BCMA− or CS1− populations by fluorescence-activated cell sorting using a FACSAria (II) sorter at the UCLA Flow Cytometry Core Facility, and the sorted polyclonal population was expanded for use in in vitro and in vivo experiments. K562 and MM.1 S cells were cultured in complete T-cell medium (RPMI-1640 (Lonza) with 10% heat-inactivated FBS (HI-FBS; Life Technologies)). HEK293T cells (ATCC) were cultured in DMEM (VWR) supplemented with 10% HI-FBS.

**Retrovirus production.** HEK 293T cells seeded in 10-cm dishes at $5.5 \times 10^6$ cells in 9 mL of DMEM + 10% HI-FBS + 20 mM HEPES (DMEM-HEPES) were transfected by linear polyethylenimine (PEI). Sixteen hours post-transfection, cells were washed with 5 mL of 1×-phosphate buffered saline without magnesium and calcium (PBS) (Lonza) and supplemented with fresh DMEM-HEPES supplemented with 10 mM sodium butyrate (Sigma–Aldrich). After 8 h, cells were washed with sterile PBS and then 8 mL of DMEM-HEPES was added. Viral supernatant was collected the following morning and cell debris was removed by filtering the viral supernatant through a 0.45 μM membrane (Corning). Six milliliters of fresh DMEM-HEPES was added to the cells following the first round of viral collection. After 24 h, a second viral harvest was performed the following day, and virus harvested from first and second batches were combined and stored at −80 °C until further use.

**Lentivirus production.** HEK 293 T cells seeded in 10-cm dishes at $3.5 \times 10^6$ cells in 9 mL DMEM + 10% HI-FBS media were transfected by linear PEI. Sixteen hours post-transfection, cells were washed with PBS and supplemented with fresh media containing 60 mM sodium butyrate (Sigma–Aldrich). Viral supernatant was collected 24 h and 48 h after media change, and cell debris was removed from the supernatant by centrifugation at $450 \times g$ for 10 min at 4 °C, followed by filtration through a 0.45 μM membrane (Corning). Viral supernatant collected 24 h after media change was mixed with ¼ volume 40% polyethylene glycol 8000 (PEG) (Amresco) in 1 × PBS and rotated overnight at 4 °C. PEG-treated virus was pelleted at $1000 \times g$ for 20 min at 4 °C, then resuspended in viral supernatant collected 48 h after media change, and finally ultracentrifuged at $51,300 \times g$ for 1 h and 35 min at 4 °C. Pellets were resuspended in 200 μL of serum-free RPMI-1640 and then incubated for 1 h at 4 °C to allow complete dissolution. Virus was then stored at −80 °C for subsequent titer and use.

**Adeno-associated virus production.** HEK 293 T cells seeded in eighteen 10-cm dishes at $3 \times 10^6$ cells in 9 mL of DMEM + 10% HI-FBS media were transfected by linear PEI. After 72 h, cells were harvested, pelleted at $1000 \times g$ for 5 min at 4 °C, then resuspended in 14.4 mL of 50 mM Tris + 150 mM NaCl (pH 8.2). The cells were lysed by undergoing three freeze/thaw cycles, then incubated at 37 °C for 1 h with benzonase (10 U/mL; EMD Millipore). The lysate was then centrifuged at $13,200 \times g$ for 10 min at room temperature. Supernatant was collected and stored at 4 °C until next step. The lysate supernatant was ultracentrifuged with iodixanol (OptiPrep; StemCell Technologies) density-gradient solutions (54%, 40%, 25%, and 15% w/v) at $76,900 \times g$ for 18 h at 4 °C. Then, 4/5 of the 40% layer and 1/5 of the 54% layer were extracted from the polyallomer Quick-seal ultracentrifuge tube (Fisher) with an 18-gauge needle (Fisher) attached to a 10-mL syringe (VWR). The collected virus fraction was diluted in an equal volume of PBS + 0.001% Tween-20, applied to an Amicon Ultra-15 (EMD Millipore, 10 kDa NMWL) column, and centrifuged at $4000 \times g$ for 20 min at 4 °C. The resulting virus fraction was diluted with PBS + 0.001% Tween-20 and centrifuged until 500 μL of the virus fraction remained in the column. Concentrated virus was stored at 4 °C for subsequent titer and use.

**Generation of CAR-expressing primary human T cells.** CD25−/CD14−/CD62L+ naïve/memory (NM), CD8+, or bulk T cells were isolated from healthy donor whole-blood obtained from the UCLA Blood and Platelet Center. CD8+ cells were also isolated using the RosetteSep Human CD8+ T Cell Enrichment Cocktail (StemCell Technologies) following manufacturer's protocols. Bulk T cells were isolated using RosetteSep Human T-cell Enrichment Cocktail (StemCell Technologies). Peripheral mononuclear blood cells (PBMCs) were isolated using Ficoll density-gradient separation, and NM T cells were subsequently isolated from PBMCs using magnetism-activated cell sorting (Miltenyi) to first deplete CD25- and CD14-expressing cells and next enrich for CD62L+ cells. Isolated T cells were stimulated with CD3/CD28 T-cell activation Dynabeads (Life Technologies) at a 1:3 bead:cell ratio. In initial screens, T cells were retrovirally transduced 48 and 72 h post stimulation. For the reduced CAR-T-cell panel, T cells were lentivirally transduced 48 h after stimulation at a multiplicity of infection of 1.5. For retrovirally and lentivirally transduced CAR-T cells, Dynabeads were removed 7 days post stimulation. For CAR-T cells with CAR integrated via homology-directed repair (HDR), Dynabeads were removed 3 days post stimulation, and T cells were nucleofected with RNP, consisting of a previously reported single-guide RNA targeting the 5′ end of exon 1 of T-cell receptor α constant (TRAC) locus[31] complexed to purified Cas9 protein. Nucleofected cells were incubated at 37 °C for 10 min, and then transduced with adeno-associated virus (AAV) at a multiplicity of infection of $3 \times 10^5$. All T cells were expanded in complete T-cell medium and fed interleukin (IL)-2 (50 U/mL; Life Technologies) and IL-15 (1 ng/mL; Miltenyi) every 2–3 days. CAR-T cells were evaluated without further cell sorting.

**Flow cytometry.** Flow cytometry in this study was performed with a MACSQuant VYB cytometer (Miltenyi Biotec). T-cells were assessed for surface presentation of epitopes using fluorophore-conjugated monoclonal antibodies for DYKDDDDK (also known as FLAG; BioLegend #637308, 1:250 dilution), HA (Miltenyi #130-092-258 or 130-120-722, 1:50 dilution), PD-1 (Miltenyi #130-117-698, 1:100 dilution), LAG-3 (eBioscience #17-2239-42 or BioLegend #369314, 1:100 dilution), CD8 (Miltenyi #130-113-164, 1:100 dilution), CD45 (BioLegend #304022, 1:100 dilution), or TCR α/β (BioLegend #306704, 1:100 dilution). EGFRt expression was measured with Erbitux (Bristol-Myers Squibb) biotinylated in-house (EZ-link Sulfo-NHS-Biotin, Pierce, 1:100 dilution). For biotin-conjugated antibodies, PE-conjugated streptavidin (Jackson ImmunoResearch #016-110-084, 1:100 dilution) was used subsequently. Tumor cells were analyzed for antigen expression with APC-conjugated anti-CS1 antibody (BioLegend #331809, 1:50 dilution) or PE-conjugated anti-BCMA antibody (BioLegend #357503, 1:50 dilution). Flow data were analyzed and gated in FlowJo (TreeStar; see Supplementary Fig. 18 for an example of gating strategy). Unless otherwise noted, data shown are drawn from biological triplicates (i.e., three distinct samples).

**Cytotoxicity assay**. Target cells ($1 \times 10^4$ cells) were seeded in a 96-well plate and coincubated with effector cells at an effector:target (E:T) ratio of 2:1 (150 μl total volume/well). Effector-cell seeding was based on CAR$^+$ T-cell count. Remaining target cells was quantified every 2 hours by GFP fluorescence imaging of target cells using IncuCyte ZOOM Live Cell Imaging System (Essen Bioscience). The amount of green fluorescence at specific time points was normalized to fluorescence at time 0 to calculate the fraction of live tumor cells remaining. Survival Kill rates were calculated by applying the log-linear model with the lm() function in R 3.5.2 software. Specifically, the fraction of live tumor cells remaining (mean of three technical replicates at each time point) were plotted on a log scale, fitted to a log-linear curve, and the absolute value of the slope of the curve was calculated to yield the kill rate. Standard error of the slope was calculated by the expression SE

$$= \sqrt{\frac{\sum (y_i - \hat{y}_i)^2 / (n-2)}{\sum (x_i - \bar{x})^2}},$$ where $y_i$ is the fraction of live tumor cells remaining at each

time point, $\hat{y}_i$ is the log-linear estimated value of the fraction of live tumor cells remaining at each time point, $x_i$ is the actual time of each time point, $\bar{x}$ is the mean of the time points, and $n$ is the number of time points.

**Proliferation assay**. Effector cells were stained with CellTrace Violet (CTV; Thermo Fisher Scientific) and coincubated with $2.5 \times 10^4$ target cells/well in a 96-well plate at an E:T ratio of 2:1, where effector-cell seeding was based on CAR$^+$ T-cell count (150 μl total volume/well). After 120 h, CTV-dilution of effector cells was quantified by flow cytometry using a MACSQuant VYB instrument (Miltenyi).

**Cytokine production**. Target cells were seeded at $5 \times 10^4$ cells/well in a 96-well plate and coincubated with effector cells at an E:T ratio of 2:1 for 24 h. Effector-cell seeding was based on CAR$^+$ T-cell count. Cytokine concentrations in the culture supernatant were measured using BD Cytometric Bead Array Human Th1/Th2 Cytokine Kit II (BD Biosciences), and data were analyzed using FCAP Array v3.0.1 Software.

**Repeated antigen challenge**. Target cells were seeded at $1.8–5 \times 10^5$ cells/well in a 48- or 24-well plate and coincubated with effector cells at an E:T ratio of 1:1 or 1:2 (1–1.5 ml total volume/well). Effector-cell seeding was based on CAR$^+$ T-cell count. Remaining target cells were quantified by flow cytometry every 2 days. Fresh target cells ($1.8–5 \times 10^5$ cells/well) were added to effector cells every 2 days after cell counting.

**In vivo xenograft studies in NOD/SCID/$\gamma_c^{-/-}$ (NSG) mice**. All in vivo experiments were approved by the UCLA Animal Research Committee (ARC). Six- to eight–week-old male and female NSG mice were bred in-house by the UCLA Department of Radiation and Oncology. Animals were housed in UCLA Division of Laboratory Animal Medicine (DLAM) facilities where temperature, humidity, and illumination were maintained according to ARC guidelines. EGFP$^+$, ffLuc-expressing MM.1 S cells ($1.5 \times 10^6$–$2 \times 10^6$) were administered to NSG mice via tail-vein injection. Upon confirmation of tumor engraftment (5–8 days post tumor cell injection), mice were treated with $0.5 \times 10^6$–$1.5 \times 10^6$ EGFRt-transduced or CAR$^+$/EGFRt$^+$ cells via tail-vein injection. In some experiments, animals were redosed 8 days later with a second injection of $1.5 \times 10^6$ T cells as noted in the text and figure captions. Tumor progression was monitored by bioluminescence imaging using an IVIS Lumina III LT Imaging System (PerkinElmer), and images were collected and analyzed using Living Image Software version 4.4 (Perkin Elmer). For combination therapy with anti-PD-1, mice were treated with 200 μg of anti-PD-1 (Ultra-LEAF, BioLegend) via intraperitoneal (i.p.) injection every 3–4 days starting one day before T-cell injection. In the experiment to evaluate anti-PD-1, animals that had been tumor-free for at least 7 weeks were rechallenged with $1.5 \times 10^6$ EGFP$^+$, ffLuc-expressing WT MM.1 S cells via tail-vein injection on day 133. At the time of sacrifice, cardiac blood and tissue (e.g., brain, spleen, and tumor mass) were collected for analysis. Prior to staining, cardiac blood was treated with Red Blood Cell Lysis Solution (Miltenyi) following manufacturer's protocol. Tissue was processed by cutting the sample finely with a surgical scissor, filtering through a 100 μm cell strainer (Corning), and then washing with PBS.

**Amplicon DNA sequencing**. Genomic DNA was isolated from $1 \times 10^6$ tumor cells using DNeasy Blood & Tissue Kit (Qiagen). BCMA and CS1 loci amplicons, with Nextera transposase adapters (Illumina) flanking each target locus, were prepared via PCR with the isolated genomic DNA. Nextera indices (Illumina) were attached to the adapters to barcode each amplicon samples via PCR. After each PCR round, amplicons were purified using AMPure XP beads (Beckman Coulter). The barcoded amplicon samples were then sent to the UCLA Technology Center for Genomics & Bioinformatics for multiplex sequencing with $2 \times 300$ paired-end configuration in a single-lane flow cell of MiSeq instrument (Illumina), and data were collected using MiSeq Control Software version 2.6.2.1 (Illumina). Fastq paired-end raw data were filtered, trimmed, and merged with DADA2 (version 1.12) on R 3.5.0 software. This work used computational and storage services associated with the Hoffman2 Shared Cluster provided by UCLA Institute for Digital Research and Education's Research Technology Group.

**Statistical analysis**. Statistical significance of in vitro results was analyzed using two-tailed, unpaired, Student's $t$-test with Bonferroni correction for multiple comparisons. Animal survival data were analyzed by log-rank analysis. All experiments were performed with biological replicates—i.e., measurements were taken from distinct samples.

**Reporting summary**. Further information on research design is available in the Nature Research Reporting Summary linked to this article.

## Data availability

All data generated during this study are available from the corresponding author upon reasonable request. The source data underlying Figs. 2, 3c–e, 4b-d, and 5, as well as Supplementary Figs. 2a, 3c, 5, 6, 7, 9, 10b, 12, 15, 16a–c, 17c–g, and 19 are provided as a Source Data file.

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

## Acknowledgements

This work used computational and storage services associated with the Hoffman2 Shared Cluster provided by UCLA Institute for Digital Research and Education's Research Technology Group. This work was supported by the Parker Institute for Cancer Immunotherapy. E.Z. was supported by the UCLA Dissertation Year Fellowship. E.N. was supported by the University of California Presidential Fellowship.

## Author contributions

Y.Y.C. and E.Z. designed the research; E.Z., E.N., V.B., U.T., and S.B.G. conducted experiments; E.Z., E.N., B.Y.J., and Y.Y.C. analyzed data. X.W. and C.E.B. provided technical advice on CS1 single-input CARs and the use of naïve/memory T cells, respectively. E.Z., E.N., and Y.Y.C. wrote the manuscript.

## Competing interests

Y.Y.C. and E.Z. declare competing financial interest in the form of a patent application whose value may be affected by the publication of this work. The other authors declare no competing interests.
