## [Peer Review File · Nature Communications]

Editorial Note: This manuscript has been previously reviewed at another journal that is not operating a transparent peer review scheme. This document only contains reviewer comments and rebuttal letters for versions considered at Nature Communications .

REVIEWERS' COMMENTS:

Reviewer #2 only left comments to the editors.

Reviewer #4 (Remarks to the Author):

The manuscript entitled 'Systematically optimized BCMA/CS1 bispecific CAR-T cell robustly control heterogeneous multiple myeloma' by Zah et al reports the construction, optimization, and in vitro and in vivo (mice) activity of a novel CAR T cell for the treatment of myeloma. The authors conclude their bispecific construct shows (1) higher CAR expression and greater antigen-stimulated proliferation than T cell that co-express individual BCMA and CS1 CARs, (2) these cells prolong survival of animals with MM tumors, and (3) combined with anti-PD-1 antibody has an uncertain long term benefit.

The authors have responded thoroughly to prior reviewers comments and have made appropriate changes that improve the manuscript.

The current BCMA directed CAR T products have had impressive response rates, but few if any long term relapse-free survivors. The report will be of interest to those in the CAR T field as well as clinicians investigating immunotherapy approaches to myeloma.

Ultimately determination of safety and efficacy of this approach further in animal models and pilot human studies will be required to evaluate the validity of this approach.